# Prediction of Possible Adverse Effects of Gestational Diabetes Mellitus on Maternal and Fetal Glomeruli by Urine and Amniotic Fluid Podocyte Degradation Products

**DOI:** 10.3390/diagnostics14242771

**Published:** 2024-12-10

**Authors:** Fatma Tanilir Cagiran, Nihal Mavral, Zercan Kali, Pinar Kirici

**Affiliations:** 1Private Clinic Obstetrics and Gynecology, 21100 Diyarbakır, Turkey; 2Private Urfa Lotus IVF Center, 63320 Urfa, Turkey; nimavral@yahoo.com; 3Gynecology and Obstetric Department, Private Gözde Hospital, 44100 Malatya, Turkey; zercankali@gmail.com; 4Gynecology and Obstetric Department, Malatya Turgut Özal Universıty, 44210 Malatya, Turkey; pinarkiricidr@hotmail.com

**Keywords:** gestational diabetes mellitus, podocyte damage, nephrin, podocalyxin, amniotic fluid

## Abstract

**Objectives:** To compare the levels of podocyte damage markers nephrin and podocalyxin in urine samples taken at the time of gestational diabetes mellitus (GDM) diagnosis and at birth. Amniotic fluid podocalyxin (pdx) and nephrin levels were also analyzed to determine whether GDM had an impact on fetal glomeruli. **Methods:** A total of 50 patients, including 24 patients diagnosed with gestational diabetes and 26 healthy pregnant women whose gestational weeks were matched, were included in the study. GDM was diagnosed with a 75 g oral glucose tolerance test (OGTT). Fresh morning urine samples from patients diagnosed with GDM were collected. The second urine sample was collected with the help of a catheter during birth. Amniotic fluid samples were taken from patients who gave birth by cesarean section. The urinary podocalyxin and nephrin levels were measured via the quantitative sandwich enzyme immunoassay. Albumin–creatinine ratio (uACR) was also calculated in urine samples. **Results:** Urinary nephrin and pdx levels on the day of GDM diagnosis were similar in the GDM and control groups. Microalbuminuria was detected in only one patient from each group at the time of GDM diagnosis. In the urine samples taken from the time of birth, pdx and nephrin levels of the GDM group were significantly higher than the control group (*p* < 0.001 for each). While microalbuminuria was detected in five patients (20.8%) at the time of birth in the GDM group, it was detected in only two patients (7.7%) in the control group. In the GDM group, a significant increase was detected between the urine pdx and nephrin levels measured at diagnosis and those measured at birth. In the control group, measurements at baseline and at birth were similar. There was no significant difference between the GDM and control groups in terms of amniotic fluid pdx and nephrin levels. A positive and significant correlation was detected between urinary pdx, nephrin, SBP, and uACR. **Conclusions:** While GDM triggers podocyte damage in maternal glomeruli, it does not cause significant change in fetal glomeruli.

## 1. Introduction

The developmental origin of health and disease (DOHAD) hypothesis holds that exposure to hyperglycemia and hyperinsulinemia during intrauterine life predisposes to the onset of some diseases later in life. Abnormal glucose tolerance, which is diagnosed or begins for the first time during pregnancy, is considered gestational diabetes mellitus (GDM) and is one of the pregnancy-specific pathologies under this hypothesis [1]. Even if glycemic control is good during pregnancy in patients with GDM, the risk of both maternal and perinatal complications is higher than the general population [2]. Mothers with gestational diabetes (GDM) and their babies are at risk for metabolic, cardiovascular, and chronic kidney diseases in the future [3,4]. Intrauterine fetal hyperglycemia and hyperinsulinemia can lead to pathology in the development of many organs related to glucose metabolism, especially the fetal pancreas and liver [2]. There are limited data on how the kidneys are affected by GDM. Clinical and experimental data have shown that maternal hyperglycemia causes a decrease in the number of nephrons and glomerular filtration rate and an increase in blood pressure and microalbuminuria [5,6]. Consistent with this, it has been reported that the fetuses of pregnant women with type 1 diabetes have a decrease in the number of nephrons and an increase in the incidence of renal dysgenesis [6,7]. The risk of hypertension, type 2 diabetes, and kidney disease due to fetal macrosomia increases in the babies of patients with GDM [8]. Although GDM causes albuminuria by impairing fetal nephron functions and reducing renal cortex volume, the effect of GDM and its components on fetal glomeruli [9] has not been fully revealed. The possible effects of metabolic pathologies on glomerular and tubules are generally determined by urine and serum parameters. Urinary activity of N-acetyl-β-d-glucosaminidase and cathepsin B, which are indicators of glomerular damage, were found to be significantly higher in newborn babies of mothers with GDM than in healthy controls [8].

Although many urinary markers are used to predict nephron and tubule damage, these markers are not more sensitive than albuminuria, nor are they cost effective. For this reason, urine levels of podocyte degradation products, which more accurately predict glomerular basal membrane damage, have begun to be analyzed [10]. Podocytes are foot-like protrusions in the epithelial structure that ensure the integrity of the glomerular basement membrane and prevent protein leakage. Nephrin and podocalyxin (pdx) are podocyte-specific proteins, and their urine levels are used as early indicators of glomerular basal membrane damage. While nephrin is more specific to the slit diaphragm in the glomeruli, pfx is localized to the podocyte plasma membrane [11,12]. Glycoprotein properties allow urine levels of both markers to be used to determine long-term glomerular damage. In good agreement with this, it has been shown that maternal urine levels of pdx and nephrin increase in high-risk pregnancies such as preeclampsia and diabetes [13]. Although podocyte-specific marker proteins in the urine of patients with GDM have been analyzed in isolated studies [13], amniotic fluid podocyte markers of babies of mothers with GDM have not yet been studied. This study was therefore planned to compare pdx and nephrin levels in spot urine samples taken from patients with GDM at the time of initial diagnosis and at birth. To determine whether exposure to GDM had an effect on fetal glomeruli, pdx and nephrin were analyzed in amniotic fluid samples taken at birth.

## 2. Materials and Methods

Data collection started after receiving the approval of the SBU Gazi Yaşargil Training and Research Hospital Clinical Research Ethics committee (Approval No: 571/17/11/2023). All participants were given detailed information about the study, and informed consent forms were obtained. Throughout the study, strict compliance with international Helsinki principles was demonstrated.

A total of 50 patients, including 24 patients diagnosed with gestational diabetes and 26 healthy pregnant women who did not have any metabolic or systemic diseases and whose gestational weeks were matched, were included in the study. Participants in the GDM group were required to have no overt diabetes or previous history of GDM as a condition for participation. GDM was diagnosed with a one-step approach [14]. A single 75-g OGTT was performed at 24–28 weeks of gestation in all participants. The cut-off values determined by the American Diabetes Association (ADA) values were used as criteria for the diagnosis of GDM [15,16]. Glucose threshold values are presented as mg/dL. Accordingly, GDM was diagnosed if any two values of the following threshold were detected: fasting glucose concentration—92 mg/dL; 1 h glucose—180 mg/dL; and 2 h glucose—152 mg/dL. Among patients with GDM, those who had a cesarean delivery were included in the study, while those who had a normal vaginal birth were excluded. While fetomaternal pathologies requiring cesarean delivery and cesarean preference as elective birth were taken into consideration, GDM was not accepted as a cesarean indication. The first maternal urine sample was taken immediately after the diagnosis of GDM, before starting diet or medical treatment. Since the patient’s mode of birth was not known in advance, the first urine sample was taken from all patients. A second urine sample was not collected from patients who gave birth normally and was not included in the study.

In patients diagnosed with GDM, glycemic control was attempted to be achieved with medical nutrition therapy and moderate physical activity. Patients were recommended to eat 3 meals balanced in terms of carbohydrates, protein, and unsaturated fat, and snacks between meals. It was recommended that they do half an hour of aerobic exercise and walking 5 days a week. The glycemic target was determined as fasting glucose < 95 mg/dL, one hour postprandial glucose level < 140 mg/dL, or two hours postprandial glucose level < 120 mg/dL. Insulin treatment was started in 4 patients whose target glucose levels could not be reached with diet and lifestyle changes [17,18,19].

The BMI of each pregnant woman was calculated by dividing the subject’s weight (kg) by the square of the subject’s height (kg/m^2^). Systolic blood pressure (SBP) and diastolic blood pressure (DBP) measurements of pregnant women in the GDM and control groups were taken in a sitting position and on the left arm, following a 10 min rest. Blood glucose and creatinine levels were measured by the hexokinase and modified Jaffe method, respectively, on the autoanalyzer.

Multiple pregnancies; patients conceived with IVF/ICSI; patients with type 2 diabetic, chronic kidney disease, or renal transplant; those who have used nephrotoxic drugs in the last three months; those with a history of kidney stones or persistent urinary infection; those with systemic disease with renal involvement; those with oligohydramnios; those who gave premature birth; and those with vaginal birth were not included in the study.

### 2.1. Urine and Amniotic Fluid Sample Collection

Fresh morning urine samples from patients diagnosed with GDM on 75 g OGTT and who had not yet started treatment for GDM were collected in sterile containers, centrifuged at 3000× *g* rpm for 3 min, and stored frozen until the day of analysis. The second urine sample was collected with the help of a catheter during cesarean delivery. Amniotic fluid samples were taken only from patients who gave birth by cesarean section. During cesarean delivery, 5–10 mL of amniotic fluid was taken with the help of a syringe before the fetal membranes were cut and they were frozen after centrifugation. Amniotic samples were not collected from patients who had normal vaginal deliveries due to potential cervicovaginal secretion contamination. Normal delivery patients were not included in the study because of the possible effect of contamination of amniotic fluid with vernix caseosa, blood, urine, and vaginal epithelium on the ELISA analysis results. In addition, the risk of possible stress/exercise proteinuria due to labor was also eliminated.

### 2.2. ELISA Analysis of Urine and Amniotic Fluid pdx and Nephrin

The frozen urine and amniotic fluid samples were thawed, and podocalyxin and nephrin levels were measured by the quantitative sandwich enzyme immunoassay using human ELISA kits (Sunred Biotechnology Company, Shanghai, China). After the absorbances of amniotic fluid and urine samples were measured at a wavelength of 450 nanometers on the Bio-Tek ELx800 (BioTek Instruments, Winooski, VT, USA), the concentrations corresponding to the absorbances were calculated in ng/mL. The pdx measurement range was 0.2–60 ng/mL, and the minimum measurable level was 0.153 ng/mL. The standard curve range of nephrin was 0.2–40 ng/mL, and the minimum measurable level was 0.16 ng/mL. While the intra-assay CV value of both kits was <10%, the inter-assay CV value was <12%.

### 2.3. Measurement of Urine Albumin–Creatinine Ratio (uACR)

Albumin–creatinine ratio (uACR) was calculated in the morning single-void fresh urine samples of participants in the GDM and control groups. Microalbuminuria was defined as uACR > 30 to 300 mg albumin/gCr excretion. Spot urine creatinine level was measured by kinetic alkaline picrate, and microalbumin level was measured by immunoturbidimetric method.

## 3. Statistical Analysis

Data were analyzed using a trail version of SPSS Statistics 27.0 for Windows (IBM Corp., Armonk, NY, USA). Graphs were drawn using GraphPad Prism 8.0 (GraphPad Software Inc., San Diego, CA, USA). Following the Shapiro–Wilk test, for variables that did not show normal distribution, the Mann–Whitney U test was used to determine statistical significance between two independent groups, and the Wilcoxon test was used to determine statistical significance between two dependent groups. For normally distributed variables, independent sample t test was used to determine statistical significance between two independent groups, and paired sample t test was used to compare dependent groups. The relationship between variables was made with Pearson or Spearman correlation tests. Fisher’s exact test was used to compare categorical variables. Data are presented as mean ± standard deviation and median (first quartile–third quartile) for continuous variables, and frequency (percentage) for categorical variables. Two-tailed *p*-values of <0.05 were considered significant.

## 4. Results

As detailed in Table 1, the median age, BMI, SBP, DBP, serum creatinine, and gestational age at birth of the patients in the GDM and control groups were similar. The fetal birth weight of the GDM group (3497.50 ± 183.97 g) was significantly higher than that of the control group (2971.92 ± 247.88 g) (*p* < 0.001). Nephrin and pdx levels measured in urine samples taken on the day of 100 g OGTT were similar in the GDM and control groups. Microalbuminuria was detected in only one patient from each group at the time of GDM diagnosis. In the urine samples taken from the time of birth, the pdx and nephrin levels of the GDM group were significantly higher than those of the control group (*p* < 0.001 for each marker). While microalbuminuria was detected in five patients (20.8%), in the urine analysis performed at the time of birth in the GDM group, it was detected in only two patients (7.7%) in the control group. No significant difference was detected between the GDM and control groups in terms of microalbuminuria prevalence on the OGTT day and at birth. In the GDM group, a significant increase was detected between the urine pdx and nephrin levels measured at diagnosis and those measured at birth (Figure 1). In the control group, measurements at baseline and at birth were similar. There was no significant difference between the GDM and control groups in terms of amniotic fluid pdx and nephrin levels (Table 2 and Figure 2).

In the GDM group, a positive and significant correlation was detected between urinary pdx, nephrin, SBP and uACR. No correlation was detected between gestational age at birth and fetal birth and urinary podocyte proteins. No relationship was detected between urinary levels of podocyte degradation products and amniotic fluid levels. No significant correlation was found between amniotic fluid podocyte degradation products and fetal and maternal demographic parameters (Table 3).

## 5. Discussion

In the context of developmental origins of health and disease, GDM can lead to exposure of the fetal kidneys to high glucose and insulin throughout pregnancy, leading to short-, medium-, and long-term renal pathologies such as decreased nephrogenesis, deterioration in glomerular functions, decreased renal cortex volume, and hypertension [8,20,21]. Similarly, GDM-mediated hyperglycemia and hyperinsulinemia may damage maternal glomeruli, causing protein leakage [13]. The damage caused by GDM to the mother’s glomeruli can be estimated based on urine albumin levels, glomerular filtration rate, and serum creatinine values. Measurements of amniotic fluid index, renal dimensions, and renal cortex volume can be performed on obstetric ultrasonography to estimate the adverse effects of GDM on fetal glomeruli. In the perinatal and neonatal periods, the effect of GDM on renal glomeruli can be examined with conventional urine analysis. However, although most of the parameters in conventional urinalysis detect severe glomerular damage, they may not be very sensitive in predicting early-stage glomerular damage.

The current study analyzed for the first time the adverse effects of GDM on maternal and fetal glomerular podocytes by measuring the urine and amniotic fluid levels of the podocyte-specific proteins pdx and nephrin. The pdx and nephrin levels measured in urine samples taken from patients diagnosed with GDM before starting any medical treatment were found to be similar to the control group. This finding is evidence that renal podocytes are healthy at the time of GDM diagnosis. The fact that microalbuminuria was detected in only one patient in the GDM and control groups and that podocyte-specific protein levels were similar in both groups is another finding that supports the supposition that the maternal systemic effects of GDM do not begin at the time of diagnosis. The detection of a significant increase in the levels of both podocyte-specific proteins in maternal urine samples taken at the time of birth, compared to both the initial measurements and the control group, is a strong finding that GDM is associated with podocyte damage.

The fact that the urinary pdx and nephrin levels of the control group did not change between diagnosis and birth was the basis for us to attribute the increase in urinary podocyte degradation products to the adverse effects of GDM. While microalbuminuria was detected in only one patient at the time of GDM diagnosis, albumin leakage was detected in five patients at birth, suggesting that podocyte damage is a process that begins simultaneously or before protein leakage. The positive correlation between uACR and urinary pdx and nephrin in the GDM group supports our idea.

Compared to the values at the time of GDM diagnosis, the detection of an approximately four-fold increase in the nephrin level and a five-fold increase in the pdx level in the urine samples taken at the time of birth supports the possibility that GDM-related metabolic changes during pregnancy cause damage to podocytes. The fact that the pdx and nephrin levels measured at birth in the control group patients were similar to the measurements at 24–28 weeks may be evidence that pregnancy-related physiological changes do not cause podocyte damage. In the GDM group, the five-fold increase in the prevalence of microalbumnuria at birth (20.83%) compared to the time of diagnosis (4.16%) suggests that protein leakage begins following or simultaneously with podocyte damage. The fact that there was no significant increase in urinary podocyte protein levels and microalbuminuria prevalence at birth and diagnosis in the control group supports that there will be no protein leakage when there is no podocyte damage. The more pronounced increase in serum markers of renal damage in patients with protein leakage supports the relationship between podocyte damage, podocyte-specific protein and albumin leakage [22]. It is a known fact that there is a decrease in the number of nephrons, an increase in protein leakage, and a decrease in the glomerular filtration rate in babies of mothers with maternal hyperglycemia [5,6]. There are isolated studies on whether GDM shows its adverse effect on maternal podocytes in the fetal glomeruli.

A limited number of studies have reported increased urine levels of N-acetyl-β-d-glucosaminidase, cathepsin B, albumin, and β2 microglubulin, which are markers of glomerular or tubular damage, in addition to a decrease in renal volume in newborns of mothers with GDM [8,9,23,24]. If GDM causes podocyte damage in maternal glomeruli that has completed maturation, it may also cause similar damage in fetal neurons in the developmental stage. However, in our study, amniotic fluid pdx and nephrin levels were similar between the babies of mothers with GDM and those in the control group. It is a paradoxical finding that hyperglycemia and hyperinsulinemia, which cause maternal podocyte damage, do not affect fetal podocytes. One of the possible explanations for this paradox may be that proteinuria and reduced nephrogenesis due to GDM become evident not in the neonatal period but in later ages [9,24]. Another possible explanation is that pdx and nephrin levels were analyzed in amniotic fluid rather than fetal urine. We did not find any study on the analysis of nephrin and pdx in amniotic fluid via the ELISA method. However, since amniotic fluid is a biological material, the majority of which is fetal urine, we thought it was a suitable material to evaluate the change in renal functions from the diagnosis of GDM to birth. A limitation of the study is that we did not analyze the urine pdx and nephrin levels of newborn babies. However, our findings are important as they show for the first time that the possible adverse effects of GDM on fetal podocytes are not evident in the neonatal period. Considering that microvascular complications of diabetes such as nephropathy occur in the long term [9,24], normal amniotic fluid podocyte-specific protein levels are an acceptable finding. Detection of maternal glomerular damage, unlike fetal glomeruli, suggests that patients with GDM are in the risk group for diabetes, undiagnosed but genetically predisposed to diabetes.

## 6. Conclusions

Although more comprehensive studies on a larger sample size and fatal urinalysis are required, our preliminary findings suggest that GDM causes maternal podocyte damage but does not cause detectable damage to fetal glomeruli in the neonatal period. Based on the fact that diabetic renal pathologies occur at later ages, normal amniotic fluid pdx and nephrin values do not guarantee that renal pathology will not develop at later ages. Finally, although microalbuminuria is the main marker routinely used to indicate glomerular damage in diabetic patients, not all glomerular damage may initially cause microalbuminuria. Analysis of urinary podocyte-specific proteins in patients with GDM and their newborns may be of critical importance in the early diagnosis of renal pathologies that are not accompanied by protein leakage or where leakage has not yet begun. However, the small sample size of this case–control study is the biggest obstacle to generalizing our results.

## Figures and Tables

**Figure 1 diagnostics-14-02771-f001:**
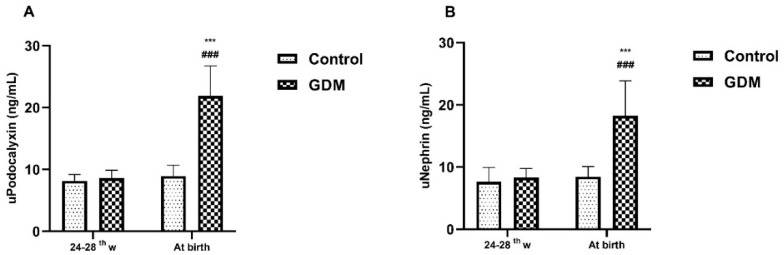
Graphical representation of the change in urinary pdx (**A**) and nephrin (**B**) levels of GDM and control groups during GDM diagnosis and at birth. Note that in the GDM group, urine levels of podocyte markers increased significantly at birth. (*** *p* < 0.001, comparison of values at GDM diagnosis and at birth, paired sample *t*-test; ### *p* < 0.001, comparison of values at birth between the GDM and the control group, independent sample *t*-test).

**Figure 2 diagnostics-14-02771-f002:**
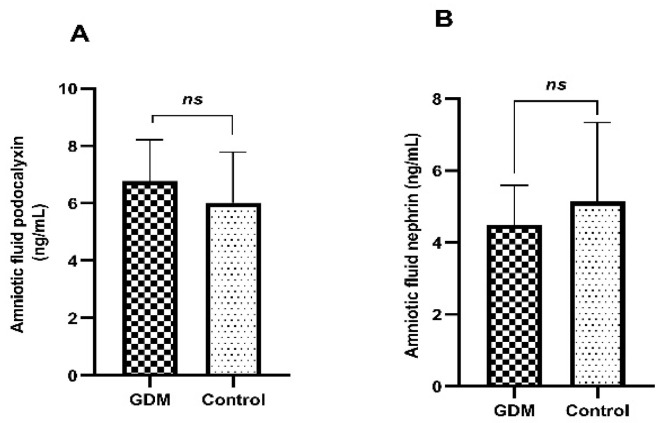
Graphical representation of the change in amniotic fluid pdx (**A**) and nephrin (**B**) levels of GDM and control groups at birth. Note that amniotic fluid levels of podocyte markers in the GDM group did not change at birth. ns depicts not significant. Independent sample *t*-test was used in both statistical comparisons.

**Table 1 diagnostics-14-02771-t001:** Demographic characteristics of pregnant women with GDM and control groups.

	GDM (*n* = 24)	Control (*n* = 26)	*p*-Value
Age (year)	28.45 ± 4.26	26.11 ± 5.17	0.089 ^a^
BMI (kg/m^2^)	25.0 (23.0–25.75)	24.0 (22.75–26.0)	0.581 ^b^
SBP (mm/Hg)	115.00 (110.0–120.0)	112.5 (110.0–120.0)	0.298 ^b^
DBP (mm/Hg)	77.5 (75.0–80.0)	75.0 (70.0–80.0)	0.162 ^b^
Creatinine (mg/dL)	0.71 ± 0.10	0.67 ± 0.12	0.351 ^a^
Gestational age at birth (weeks)	37 (37–38)	37 (37–39)	0.873 ^b^
Fetal birth weight (gr)	3497.50 ± 183.97	2971.92 ± 247.88	<0.001 ^a^

GDM: Gestational diabetes mellitus; BMI: Body mass index; DBP: Diastolic blood pressure; SBP: Systolic blood pressure. Data are given as mean ± standard deviation or median (first quartile–third quartile). ^a^ Independent sample *t*-test; ^b^ Mann–Whitney U test.

**Table 2 diagnostics-14-02771-t002:** Comparison of urinary and amniotic fluid podocyte damage markers and albuminuria frequency in GDM and control groups.

	GDM (*n* = 24)	Control (*n* = 26)	*p*-Value
Maternal urine podocyte marker levels on the day of 75 g OGTT			
Urinary nephrin (ng/mL)	8.30 ± 1.45	7.64 ± 2.29	0.227 ^a^
Urinary podocalyxin (ng/mL)	8.63 ± 1.24	8.18 ± 0.99	0.164 ^a^
Microalbuminuria (uACR > 30–300 mg/gCr)	1 (4.12%)	1 (3.8%)	1.000 ^c^
Podocyte-specific marker levels in urine samples taken at birth			
Urinary nephrin (ng/mL)	18.23 ± 5.63	8.45 ± 1.62	<0.001 ^a^
Urinary podocalyxin (ng/mL)	21.90 ± 4.83	8.95 ± 1.76	<0.001 ^a^
Microalbuminuria (uACR > 30–300 mg/gCr)	5 (20.8%)	2 (7.7%)	0.239 ^c^
Amniotic fluid podocyte-specific marker levels			
Amniotic fluid nephrin (ng/mL)	4.49 ± 1.10	5.14 ± 2.19	0.187 ^a^
Amniotic fluid podocalyxin (ng/mL)	6.83 (5.60–7.92)	6.65 (4.49–7.48)	0.177 ^b^

Data are given as median (first quartile–third quartile) for continuous variables and as frequency (percentage) for categorical variables.^ a^ Student *t*-test; ^b^ Mann–Whitney U test; ^c^ Fisher’s exact test.

**Table 3 diagnostics-14-02771-t003:** Correlations between podocytes proteins maternal and fetal demographic variables in GDM.

		Urinary pdx	Urinary Nephrin
Age (year)	*r*	0.061	0.045
	*p*	0.074	0.662
BMI (kg/m^2^)	*r*	0.439	0.309
	*p*	0.035	0.061
SBP (mm/Hg)	*r*	0.411	0.398
	*p*	0.019	0.013
DBP (mm/Hg)	*r*	0.233	0.023
	*p*	0.081	0.048
uACR (mg/gCr)	*r*	0.678	0.643
	*p*	0.001	0.001
Creatinine (mg/dL)	*r*	0.143	0.034
	*p*	0.322	0.455
Gestational age at birth	*r*	0.322	0.131
	*p*	0.399	0.430
Fetal birth weight	*r*	0.0487	0.244
	*p*	0.410	0.355

*r*: correlation coefficient; *p*: *p*-Value.

## Data Availability

All data generated or analyzed during this study are included in this article.

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
