# Peer review of "Prediction of Possible Adverse Effects of Gestational Diabetes Mellitus on Maternal and Fetal Glomeruli by Urine and Amniotic Fluid Podocyte Degradation Products"

_diagnostics, 2024, doi:10.3390/diagnostics14242771_

Round 1

Reviewer 1 Report

Comments and Suggestions for Authors

Thankyou for the invitation to review this manuscript, it was well conducted research and well written manuscript.

Just a couple of comments:

1. Title: Should the title read "Prediction of Possible Adverse Effects of GDM on Maternal and Fetal Glomeruli by Urine and amniotic Fluid Podocyte Degradation Products" instead of ...."Renal Glomeruli"?

2. There is an abbreviation in the abstract for podocalyxin which needs to be replaced by the full word.

3. Line 63 is awkward and should be rephrased

4. Why were only women whose babies were delivered by caesarian section included? Need to expand on this decision.

5. How were the urinary albumin and creatinine measured? What methods?

6. The first line of the discussion should be changed to read "In the context of Developmental origins of health and disease....." 

7. "Finding that supports GDM causes podocyte...." should be changes to "Finding that GDM is associated with podocyte damage"

Thankyou

Author Response

Q1. Title: Should the title read "Prediction of Possible Adverse Effects of GDM on Maternal and Fetal Glomeruli by Urine and amniotic Fluid Podocyte Degradation Products" instead of ...."Renal Glomeruli"?

R1: Title was changed as follow “Prediction of Possible Adverse Effects of GDM on Maternal and Fetal Glomeruli by Urine and Amniotic Fluid Podocyte Degradation Products”

Q2. There is an abbreviation in the abstract for podocalyxin which needs to be replaced by the full word.

R2: Full word of “podocalyxin” was provided before abbreviation.

Q3. Line 63 is awkward and should be rephrased.

R3: Line 63 was replaced as follow “Podocytes are foot-like protrusions that maintain the integrity of the glomerular basement membrane and prevent protein leakage”

Q4. Why were only women whose babies were delivered by caesarian section included? Need to expand on this decision.

R4: Normal delivery patients were not included in the study because of the possible effect of contamination of amniotic fluid with vernix caseosa, blood, urine and vaginal epithelium on ELISA analysis results. In addition, the risk of possible stress/exercise proteinuria due to labor was also eliminated.

Q5. How were the urinary albumin and creatinine measured? What methods?

R5: Urine creatinine level was measured by kinetic alkaline picrate, and microalbumin level was measured by immunoturbidimetric method.

Q6. The first line of the discussion should be changed to read "In the context of Developmental origins of health and disease....."  It was changed.

Q7. "Finding that supports GDM causes podocyte...." should be changes to "Finding that GDM is associated with podocyte damage"…It was changed as you suggested.

Reviewer 2 Report

Comments and Suggestions for Authors

This is a very interesting paper.

OGTT must be explicit in the abstract.  It would be necessary to inform which institution has the license to use SPSS..  In the footer of Table 1 it should be indicated which statistical test is used for the significances..  The same in the rest of the tables and figures.  In the conclusions, greater emphasis should be placed on the small sample size that compromises the conclusions of a case-control design such as this one.

Author Response

Q1. OGTT must be explicit in the abstract.  It was provided.

Q2. It would be necessary to inform which institution has the license to use SPSS. A trial version of SPSS was used.

Q3. In the footer of Table 1 it should be indicated which statistical test is used for the significances. The same in the rest of the tables and figures. It was provided in the Tables and Figures.

Q4. In the conclusions, greater emphasis should be placed on the small sample size that compromises the conclusions of a case-control design such as this one.

The Conclusion section highlighted that small sample size was an obstacle to generalization of results.